# Characterization of the innate immune response to *Streptococcus pneumoniae* infection in zebrafish

Anni K. Saralahti[1], Sanna-Kaisa E. Harjula[1], Tommi Rantapero[2]☉¤a, Meri I. E. Uusi-Mäkelä[1]☉, Mikko Kaasinen[1], Maiju Junno[1], Hannaleena Piippo[1¤b], Matti Nykter[2,3], Olli Lohi[4], Samuli Rounioja[5], Mataleena Parikka[6], Mika Rämet[1,7]*

**1** Laboratory of Experimental Immunology, Faculty of Medicine and Health Technology, Tampere University, Tampere, Finland, **2** Laboratory of Computational Biology, Faculty of Medicine and Health Technology, Tampere University, Tampere, Finland, **3** Foundation for the Finnish Cancer Institute, Helsinki, Finland, **4** Tampere Center for Child, Adolescent and Maternal Health Research, Faculty of Medicine and Health Technology, Tampere University and Tays Cancer Center, Tampere University Hospital, Tampere, Finland, **5** Fimlab Laboratories, Tampere, Finland, **6** Laboratory of Infection Biology, Faculty of Medicine and Health Technology, Tampere University, Tampere, Finland, **7** FVR–Finnish Vaccine Research, Tampere, Finland

☉ These authors contributed equally to this work.
¤a Current address: Genevia Technologies Ltd, Tampere, Finland
¤b Current address: Laboratory of Infection Biology, Faculty of Medicine and Health Technology, Tampere University, Tampere, Finland
* mika.ramet@tuni.fi

**Data Availability Statement:** The RNA sequencing data presented in this publication has been deposited in NCBI's Gene Expression Omnibus under GEO Series accession number GSE112433.

## Abstract

*Streptococcus pneumoniae* (pneumococcus) is one of the most frequent causes of pneumonia, sepsis and meningitis in humans, and an important cause of mortality among children and the elderly. We have previously reported the suitability of the zebrafish (*Danio rerio*) larval model for the study of the host-pathogen interactions in pneumococcal infection. In the present study, we characterized the zebrafish innate immune response to pneumococcus in detail through a whole-genome level transcriptome analysis and revealed a well-conserved response to this human pathogen in challenged larvae. In addition, to gain understanding of the genetic factors associated with the increased risk for severe pneumococcal infection in humans, we carried out a medium-scale forward genetic screen in zebrafish. In the screen, we identified a mutant fish line which showed compromised resistance to pneumococcus in the septic larval infection model. The transcriptome analysis of the mutant zebrafish larvae revealed deficient expression of a gene homologous for human *C-reactive protein* (*CRP*). Furthermore, knockout of one of the six zebrafish *crp* genes by CRISPR-Cas9 mutagenesis predisposed zebrafish larvae to a more severe pneumococcal infection, and the phenotype was further augmented by concomitant knockdown of a gene for another Crp isoform. This suggests a conserved function of C-reactive protein in anti-pneumococcal immunity in zebrafish. Altogether, this study highlights the similarity of the host response to pneumococcus in zebrafish and humans, gives evidence of the conserved role of C-reactive protein in the defense against pneumococcus, and suggests novel host genes associated with pneumococcal infection.

All other relevant data are within this article or its
Supporting information files.

**Funding:** This work was supported by Sigrid
Juselius Foundation; http://sigridjuselius.fi/en/
(MR), the Competitive State Research Financing of
the Expert Responsibility area of Tampere
University Hospital (MR), the Competitive State
Research Financing of the Expert Responsibility
Area of Oulu University Hospital (MR), the
Tampere Tuberculosis Foundation; http://www.
tuberkuloosisaatio.fi/ (MR, S-KH), the Finnish
Cultural Foundation, the Central Fund; http://www.
skr.fi/en/central-fund-grants (S-KH), the Finnish
Cultural Foundation, the Pirkanmaa Regional Fund;
http://www.skr.fi/en/finnish-cultural-foundation/
regional-funds/pirkanmaa-regional-fund (AS), The
Doctoral Programme in Medicine and Life
Sciences, Tampere University (AS, MU, TR), the
Emil Aaltonen Foundation; https://emilaaltonen.fi/
apurahat/in-english/ (AS, S-KH), Foundation of the
Finnish Anti-Tuberculosis Association; https://
www.tb-foundation.org/ (S-KH), the Scientific
Foundation of the City of Tampere; https://www.
tampere.fi/tampereen-kaupunki/yhteystiedot-ja-
asiointi/avustukset/apurahat/tiederahasto.html (AS,
S-KH), the University of Tampere Foundation;
http://www.uta.fi/tukisaatio/english/ (S-KH), the
Maud Kuistila Memorial Foundation; http://
mkmsaatio.fi/en/the-maud-kuistila-memorial-
foundation/ (AS, S-KH), the Väinö and Laina Kivi
Foundation; http://www.foundationweb.net/kivi/ (S-
KH), the Finnish Concordia Fund; http://www.
konkordia-liitto.com/english/ (S-KH), and the Oskar
Öflund Foundation; http://oskaroflund.fi/?lang=fi
(AS). The funders had no role in study design, data
collection and analysis, decision to publish, or
preparation of the manuscript.

**Competing interests:** The authors have declared
that no competing interests exist.

## Author summary

The innate immune system plays an important role in the recognition and activation of the phagocytic killing of *Streptococcus pneumoniae* (pneumococcus), and defects in these mechanisms are suggested to predispose individuals to a more severe infection. Due to their amenability to genetic studies and the similarities between the zebrafish and the human innate immune responses, zebrafish are good models for studying the genetic susceptibility to pneumococcal infection. Here we show that pneumococcus activates the same innate immune responses in zebrafish larvae as in humans, including a common inflammatory response, complement-mediated immunity, and phagocytic clearance. We also propose that, as in humans, the C-reactive protein (CRP) plays a role in the innate immune response to pneumococcus in zebrafish larvae, as zebrafish with a mutated version of the gene homologous to human *CRP* develop more severe sepsis, and present increased mortality in our model of systemic pneumococcal infection. Moreover, by using the zebrafish model, we reveal novel host genes which may affect the infection susceptibility also in humans and thus, provide new insights into the innate immune response to pneumococcus as well as potential targets for future drug development.

## Introduction

*Streptococcus pneumoniae* (pneumococcus) is a Gram-positive opportunistic bacterium and an important human pathogen. Typically, pneumococcus is an asymptomatic inhabitant of the human nasopharynx, but in favorable conditions the colonization may lead to the outbreak of an infection. Pneumococcus is capable of causing a wide variety of infectious diseases, most typically in the respiratory tract. It is one of the leading causes of community-acquired pneumonia but also a common cause of milder upper respiratory tract infections, and acute otitis media [1,2]. Occasionally, pneumonia may manifest as an invasive disease, and lead to the outbreak of life-threatening bacteremia and meningitis [1,2]. Due to the high incidence of pneumococcal infections, this pathogen is a major cause of morbidity especially in children and the elderly, and accounted for over a million deaths globally in 2016 [3]. While the global use of pneumococcal conjugate vaccines has reduced the colonization, pneumonia, acute otitis media, and invasive disease caused by the vaccine serotypes, the increased prevalence of non-vaccine serotypes and the high incidence of antibiotic resistant isolates of *S. pneumoniae* retain this pathogen as a major health burden worldwide [4–6].

While the carriage rate of pneumococcus can be as high as 60% of the population, colonization leads to infection only in a minority of carriers [7]. In addition, the outcome of pneumococcal infection, ranging from local to invasive, is poorly predictable. The progression and outcome of pneumococcal disease is affected by both host and bacterial factors. Importantly, pneumococcal isolates exhibit notable diversity at the level of capsular polysaccharides, which is the basis for the division of pneumococcus into at least 100 serotypes [8]. Epidemiological studies have revealed that these serotypes vary in global distribution and target group as well as in the rate of colonization, invasive capacity, and the risk of fatal infection [9–11]. Besides the capsule, the virulence of pneumococcal strains is affected by the variation in the other virulence factors of pneumococcus, such as PspA, PspC, pneumolysin, and the pilus, to name a few [12–17]. On the host side of view, the major risk factors for severe pneumococcal infection include age under 2 years or over 65 years, underlying chronic diseases, and an immunocompromised state [18,19]. In particular, patients incapable of producing polysaccharide-specific

antibodies are prone to severe pneumococcal infection, indicating that the anticapsular antibody response is the most important host defense mechanism in pneumococcal infections [20]. However, data gathered from murine models and human polymorphisms have also highlighted the importance of the innate immune system in the protection against pneumococcus, by means of the activation of adaptive responses, but also, by providing natural protection through the promotion of phagocytic clearance. Numerous genetic association studies have proposed linkage between polymorphism in innate immune response genes and the susceptibility to pneumococcal infection (reviewed in Kloek et al., 2019) [21]. These genes include, for example, genes coding for Toll- and NOD-like receptors and their signaling pathways (e.g. TLR2, TLR4, MyD88, TIRAP), cytokines (e.g., IL6, IL10), as well as complement components and acute phase response proteins (e.g. MBL2, C3, CRP, C5) [e.g. 20,22–27]. A recent meta-analysis also confirmed two susceptibility polymorphisms, one in *MBL2* and one in *CD14* [21]. However, in general, individual association studies have often given conflicting results emphasizing the complexity of the genetic basis of pneumococcal disease and, therefore, further studies on the factors affecting disease susceptibility and severity are needed.

Along with the studies on genetic variation in humans, transcriptome analyses and genetic screens conducted in animal models are useful in providing insights into the genetic basis of host response to infection at a genome-wide level. To study the details of the immune response to pneumococcal infections and to identify novel susceptibility factors, we employed the previously established zebrafish (*Danio rerio*) model of systemic pneumococcal infection [28,29]. Zebrafish are small teleosts that are frequently used in the study of immune function and infectious diseases [30]. The value of zebrafish in the study of host-pathogen interaction is emphasized by the impressive number of infection models established in the past twenty years, including the models for *Mycobacterium marinum* [31,32], *Salmonella typhimurium* [33], *Edwardsiella tarda* [34], *Staphylococcus aureus* [35], *Listeria monocytogenes* [36], and several streptococcal species [37–39]. Importantly, the zebrafish immune system is fairly similar to the human immune system, comprising both the innate and adaptive arms in adult fish [40]. On the other hand, until 4 weeks post fertilization, zebrafish larvae possess only the innate immune system, providing a unique opportunity to study the vertebrate innate responses to pathogens without the influence of the adaptive components [41]. As is typical for a vertebrate, the zebrafish innate immune system is well conserved, and contains the counterparts for most human cell types, receptors, cytokines, chemokines, and complement components [42–44]. As an additional benefit, zebrafish are highly amenable to genetic manipulation, and due to their small size, high fecundity, and easy maintenance, are especially well suited for larger-scale experiments, such as forward genetic screens [45].

In our previous studies, we have shown that the basic pathogenic mechanisms as well as the host responses to pneumococcus are conserved between zebrafish and humans [28,39]. In the present study, we aimed to characterize the zebrafish innate immune response to pneumococcus in more detail and at the same time, identify novel genetic factors affecting susceptibility to pneumococcal infection. To identify the host genes expressed during the early response to pneumococcus, we conducted a whole-genome level transcriptome analysis for the infected zebrafish larvae. Moreover, we carried out a forward genetic screen for the host genes associated with increased susceptibility to severe pneumococcal infection in zebrafish larvae. As a result, we found that similar mechanisms of innate immunity, such as complement-mediated clearance, are important in pneumococcal infection in zebrafish as in mammals, which further validates the use of this model in the evaluation of the host-pathogen interactions during pneumococcal infection. We also report a set of up- and downregulated genes and non-coding RNAs not previously associated with the innate immune response to pneumococcus. Finally, utilizing both forward and reverse genetics, we show that zebrafish that have defects either in

the production or the structure of C-reactive protein (Crp) are more susceptible to a severe pneumococcal infection than wild type zebrafish.

## Results

### Changes in zebrafish gene expression upon pneumococcal infection

In our previous studies we reported that the intravenous injection of pneumococcus induces a common inflammatory response in zebrafish larvae and that the circulating bacteria are cleared by the phagocytosing leukocytes [28]. In the present study, we further characterized the innate immune response to pneumococcus in zebrafish larvae through a whole-genome level transcriptome analysis. For the analysis, 2 dpf (days post fertilization) zebrafish larvae were infected with 466–542 colony forming units (cfu) of TIGR4 (T4) strain of *S. pneumoniae*, and the total RNA was collected at 18 hours post infection (hpi) from the pools of ten challenged or unchallenged larvae. According to our previous studies in zebrafish larvae [28], a time point of 18 hpi represents a stage of pneumococcal infection at which the bacteria are replicating rapidly, the host innate immune responses have been activated and the signs of infection (e.g. lack of movement) appear, and was therefore chosen for the analysis. The transcriptome analysis was carried out from three biological replicates by RNA sequencing. The RNA sequencing yielded 21–28 million reads per library and approximately 90% were successfully mapped to the transcript database based on the GRCz10 reference genome. To analyze the differentially expressed genes between *S. pneumoniae* challenged and unchallenged larvae, a fold change of ≥3 was considered significant induction or reduction. In addition, only genes with a mean normalized read count of ≥20 (calculated from the normalized read counts of three biological replicates) after infection were included in the analysis. Finally, two genes with significant variation in their read counts (normalized read counts of 0, 0, and 67 for gene *si:ch73-44m9.5* and 0, 0, and 167 for gene *BX640576.2*) were excluded from the analysis.

With the cutoffs described above, the transcriptome analysis revealed 132 differentially expressed genes in the larvae infected with *S. pneumoniae* with 81 of them upregulated and 51 downregulated (**Table 1** and **S1**–**S4**; the complete expression data are available at Gene Expression Omnibus [46] under accession number GSE112433). The differentially expressed protein coding genes were subjected to gene ontology (GO) analysis using the DAVID 6.7 Functional annotation tool [47,48]. According to the analysis, the most enriched biological processes reaching the p-value of <0.05 were proteolysis, defense response to bacterium, complement activation, chitin metabolic process, and defense response to Gram-negative bacterium. However, the DAVID, as well as other tested annotation tools (GOrilla and the Gene Ontology Consortium enrichment analysis [49,50]), only recognized approximately 30% of the submitted genes, and thus gave a distorted view of the enriched processes. Therefore, we manually divided the upregulated protein coding genes into the biological functions according to the available GO data (of zebrafish genes and their mouse and human orthologs) from the Ensembl genome browser versions 91 and 95 [51], The Zebrafish Information Network (ZFIN) [52], the NCBI Gene database [53], and literature. Of the 61 protein coding genes induced upon pneumococcal infection, 29 genes have previously been associated with an immune response to bacterial infection, emphasizing the activation of the zebrafish defense mechanisms against pneumococcus (**Fig 1** and **Table 1**). The other induced protein coding genes were loosely categorized into metabolic processes (12 genes), other processes (7 genes), and unknown processes (13 genes) (**Fig 1** and **S1 Table**). Among the downregulated protein coding genes with available annotation data, the most enriched functional group was reproduction and development (8 genes), while one gene was associated with metabolic processes and six genes were categorized in the group of other processes (**S2 Table**). In contrast, for a

**Table 1. Immune system–related genes induced in pneumococcal infection in zebrafish.**

| Gene symbol | Gene name | Ensembl gene ID | Fold change |
|---|---|---|---|
| **complement system** | | | |
| c3a.2 | complement component c3a, duplicate 2 | ENSDARG00000087359 | 28.9 |
| c3a.3 | complement component c3a, duplicate 3 | ENSDARG00000052207 | 8.9 |
| c1r | complement component 1, r subcomponent | ENSDARG00000100248 | 6.5 |
| cfb | complement factor B | ENSDARG00000055278 | 4.5 |
| si:ch1073-280e3.1 | (a putative counterpart for human C2) | ENSDARG00000019772 | 4.3 |
| c3a.1 | complement component c3a, duplicate 1 | ENSDARG00000012694 | 4.0 |
| cfhl5 | complement factor H like 5 | ENSDARG00000105052 | 3.9 |
| c4b | complement 4B (Chido blood group) | ENSDARG00000038424 | 3.8 |
| c3a.6 | complement component c3a, duplicate 6 | ENSDARG00000043719 | 3.0 |
| serping1 | serpin peptidase inhibitor, clade G (C1 inhibitor), member 1 | ENSDARG00000058053 | 3.0 |
| **cell migration and chemotaxis** | | | |
| CABZ01001434.1 | (a predicted chemokine) | ENSDARG00000098602 | 8.2 |
| ccl34a.4 | chemokine (C-C motif) ligand 34a, duplicate 4 | ENSDARG00000090873 | 5.3 |
| ccl19b | chemokine (C-C motif) ligand 19b | ENSDARG00000039351 | 3.1 |
| selplg | selectin P ligand | ENSDARG00000105023 | 3.1 |
| **antimicrobial peptides** | | | |
| hamp | hepcidin antimicrobial peptide | ENSDARG00000102175 | 3.9 |
| chia.2 | chitinase, acidic.2 | ENSDARG00000099185 | 3.8 |
| rnasel3 | ribonuclease like 3 | ENSDARG00000036171 | 3.7 |
| chia.1 | chitinase, acidic.1 | ENSDARG00000100635 | 3.5 |
| **immune signaling and regulation** | | | |
| lrrc66 | leucine rich repeat containing 66 | ENSDARG00000093370 | 6.2 |
| apoa4b.2 | apolipoprotein A-IV b, tandem duplicate 2 | ENSDARG00000094929 | 5.7 |
| ly6m2 | lymphocyte antigen 6 family member M2 | ENSDARG00000104775 | 3.8 |
| gpr84 | G protein-coupled receptor 84 | ENSDARG00000077308 | 3.7 |
| **acute phase response** | | | |
| crp2 | C-reactive protein, pentraxin-related | ENSDARG00000056498 | 14.8 |
| itln3 | intelectin 3 | ENSDARG00000003523 | 7.7 |
| crp3 | C-reactive protein 3 | ENSDARG00000042613 | 3.4 |
| **macrophage function** | | | |
| mfap4.5 | microfibril associated protein 4, duplicate 5 | ENSDARG00000090557 | 6.5 |
| acod1 | aconitate decarboxylase 1 | ENSDARG00000069844 | 4.6 |
| mfap4.2 | microfibril associated protein 4, duplicate 2 | ENSDARG00000089667 | 3.0 |

The table represents the fold change in expression in *S. pneumoniae* challenged larvae compared to unchallenged larvae at 18 hpi. The data comprise three biological replicates and the fold change was calculated using the DEseq2 tool. The table includes only the genes with a mean read count of $\geq 20$ after infection, and the genes induced by at least 3.0-fold compared to unchallenged larvae.

total of 18 downregulated protein coding genes no functional prediction was found. In addition to the protein coding genes, 20 noncoding RNAs were upregulated and 18 downregulated upon pneumococcal infection (**S3 and S4 Tables**).

## The transcriptome analysis indicates an innate immune response to pneumococcus in zebrafish, with conserved pathways

Our transcriptome data provide evidence for the activation of an innate immune response to pneumococcal infection at the level of gene expression. To characterize this early response in

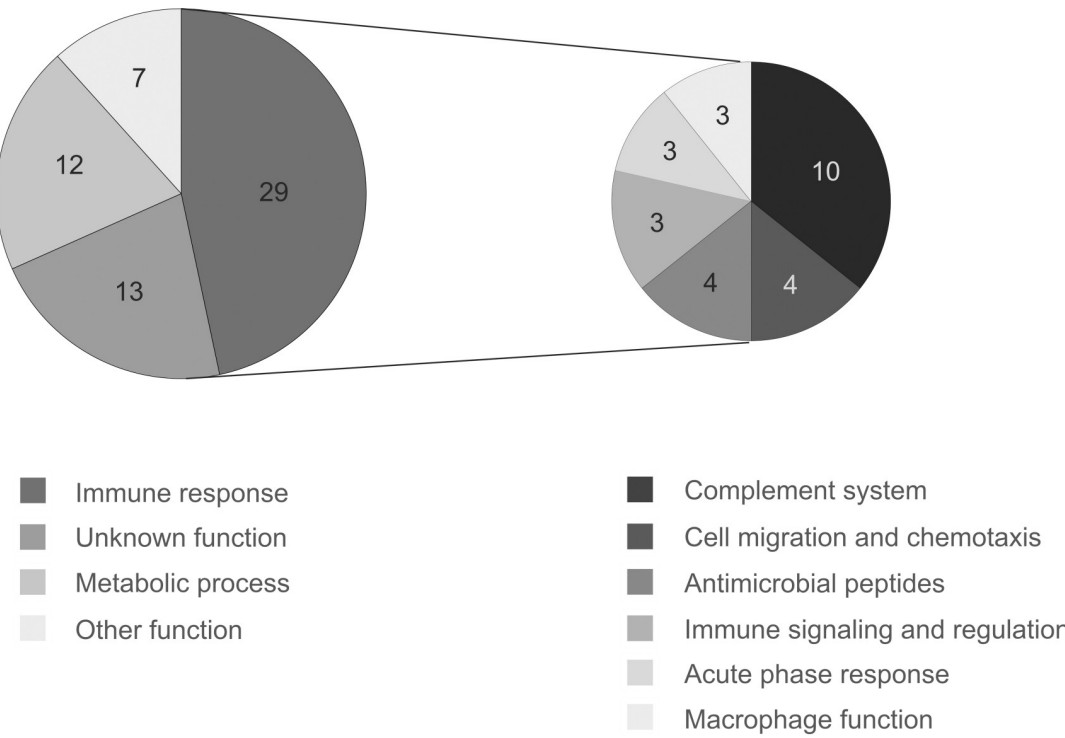

**Fig 1. Functional categories of the pneumococcus-responsive genes in zebrafish larvae.** The genes were divided into functional groups based on the available gene ontology data obtained from the Ensembl genome browser versions 91 and 95, The Zebrafish Information Network (ZFIN) and literature (pie chart on the left). Immune response genes were further divided into subcategories (pie chart on the right). The numbers in the sectors indicate the number of genes in each functional group.

more detail, the pneumococcus-responsive genes associated with innate immunity were further divided into the functional groups of complement system (10 genes), cell migration and chemotaxis (4 genes), antimicrobial peptides (4 genes), immune signaling and regulation (3 genes), acute phase response (3 genes), and macrophage function (3 genes) (**Fig 1 and Table 1**). Of note, the genes associated with the complement system were particularly enriched in our transcriptome data, suggesting an important role for complement-mediated immunity in the defense against pneumococcus in zebrafish larvae. Among the enriched complement-related genes were those coding for four variants of C3 (*c3a.1 c3a.2, c3a.3, c3a.6*), a putative counterpart for human C2 (*si:ch1073-280e3.1*), C4b (*c4b*), C1r (*c1r*), complement factor B (*cfb*), and two negative regulators, complement factor H like 5 (*cfhl5*) and C1 inhibitor (*serping1*) (**Fig 2**). In addition, two zebrafish homologs for the human gene encoding C-reactive protein (*crp2* and *crp3*, ENSDARG00000056498 and ENSDARG00000042613, respectively) were induced upon infection in zebrafish larvae. CRP in mammals is known to recognize *S. pneumoniae* and to have multiple functions in antipneumococcal response including the activation of the complement system and phagocytosis [54].

Besides the induction of Crp-encoding genes and complement genes, the early response to pneumococcus in zebrafish was characterized by the expression of other genes previously associated with innate immune response to pathogens (**Table 1**). These included the genes for antimicrobial peptide hepcidin (*hamp*) [55] and for one of the Ribonuclease A superfamily members (*ribonuclease-like 3, rnasel3*) which have been shown to have antibacterial, angiogenic and neurotrophic activity in zebrafish [56,57]. Similarly, the expression of two zebrafish chitinase genes *chia.1* and *chia.2*, shown to have antimicrobial activity but also other

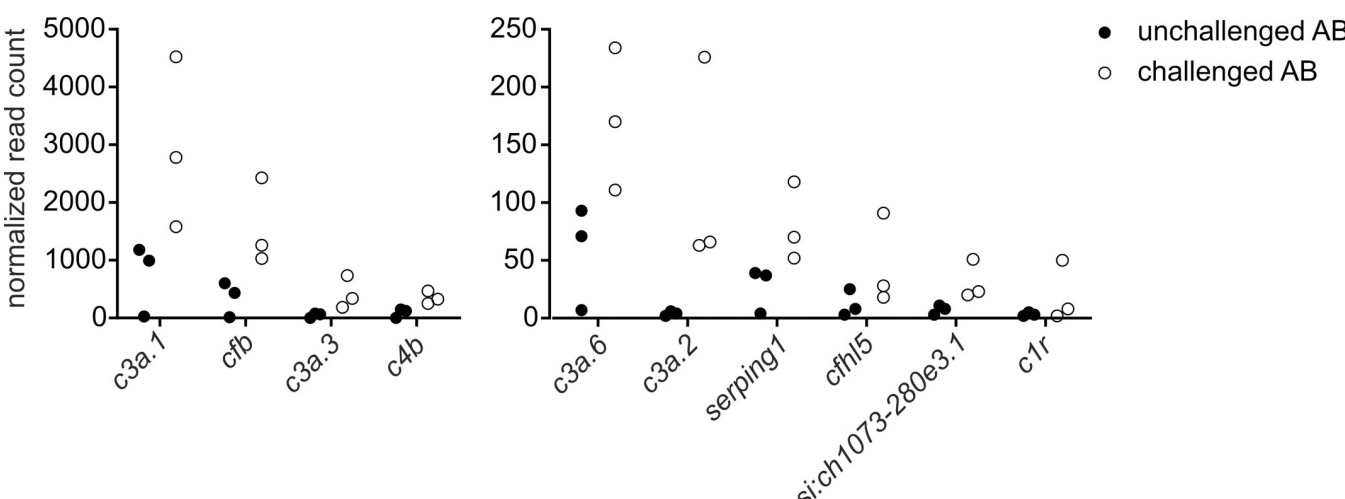

**Fig 2. The induced complement -related genes in pneumococcal infection at 18 hpi.** Normalized read counts in wild type zebrafish larvae injected with KCl (unchallenged AB) or ~500 cfu of *S. pneumoniae* (challenged AB). The genes which had normalized read counts of ≥20 after infection and were induced by ≥3-fold are shown. Each dot represents the normalized read count of a sample of 10 larvae. The differential expressions did not reach statistical significance.

immunoregulatory functions in fish and mammals [58–62], were upregulated. The zebrafish *itln3*, coding for a member of the well-conserved lectin family which serves multiple functions in pathogen recognition and the regulation of inflammation [63–66], was also among the most induced genes at 18 hpi. The category of the immune signaling and regulation, on the other hand, included the lipid transport gene *apoa4b.3*, the G-protein coupled receptor gene *gpr84* [67] and a member of lymphocyte antigen 6 family (*ly6m2*). In addition, *lrrc66*, an as yet functionally uncharacterized gene coding for a protein with a leucine-rich repeats (LRR)-containing domain, which in many pattern-recognition receptors mediates the receptor-pathogen interaction [68], was induced.

The upregulation of the genes associated with leukocyte activation, migration and phagocytic functions were also detected from the transcriptome data, supporting their important role in the clearance of pneumococcus (**Table 1**). These included genes for two chemokines (*ccl34a.4* and *ccl19b*), a predicted chemokine (*CABZ01001434.1*) and a selectin P ligand (*selpgl*) which in mammals promotes leukocyte recruitment and which shows conserved selectin binding in zebrafish [69]. Also, a macrophage-specific gene *aconitate decarboxylase 1*, *acod1* (also known as *immunoresponsive 1 homolog*, *irg1*), which is known to be highly expressed under bacterial infections and to be important for the production of reactive oxygen species by macrophages [70,71], was among the most induced genes in pneumococcal infection. Moreover, consistent with the previous studies on *Staphylococcus epidermis*-responsive genes in zebrafish, two duplicates of the gene *microfibril associated protein 4* (*mfap4.5* and *mfap4.2*), a well-known macrophage marker shown to promote macrophage differentiation during hematopoiesis in zebrafish [72,73], were also upregulated upon infection. Of note, while not quite reaching the cutoff levels set for the induced genes, the induction of the genes for common pro-inflammatory cytokines Il1b (4.4-fold induction) and Il6 (9.9-fold induction) was also evident, with mean normalized read counts of 14 and 4.6 in challenged larvae, and 3 and 0.3 in unchallenged larvae, respectively.

## Identification of novel pneumococcus-responsive genes

Besides the genes related to the immune response, upregulation of a set of protein coding genes associated with metabolic processes (12 genes) and other functions (7 genes), was

detected in the transcriptome analysis (**S1 Table**). While their relation to immune response is yet to be unraveled, these genes represent potential novel factors with a role in the early response to pneumococcus. For example, the functional group of metabolic process contained 6 (predicted) serine proteases (*ctrl*, *ctrb1*, *ela3l*, *prss59.1*, *prss59.2*, *ctrb.3*), a group of proteolytic enzymes participating in various biological processes including the immune response [74,75]. In addition, the analysis also revealed multiple upregulated protein coding genes and non-coding RNAs (**S1 and S3 Tables**) with unknown function, also serving as candidates for further functional characterization.

## The forward genetic screen identifies a fish line with compromised resistance to pneumococcal infection

To identify novel genes affecting the innate immune response to pneumococcal infection, we carried out a forward genetic screen in zebrafish utilizing a transposon-based mutagenesis method created by Clark et al. (2011) [76]. In the screen, we generated 126 insertional mutant zebrafish lines, and screened them in F3-F4 generations for altered susceptibility to pneumococcal infection by survival assays. All the mutant lines chosen for the assays survived normally in the laboratory conditions and did not show any defects in morphology, development, or well-being. Zebrafish larvae showing GFP expression as a sign of the presence of one or more randomly located mutations in their genome, were selected for the assays. In the survival assays, 2 dpf zebrafish larvae from each mutant line were infected with *S. pneumoniae* and their survival was compared to the AB wild type line and the other mutant lines in the same experiment. A total of 58 survival experiments were conducted (bacterial doses ranging from 30–570 cfu between experiments). In each experiment, a wild type control and at least three mutant lines were included to distinguish hypersusceptible lines. In the initial screening, about 10% of the lines appeared to have altered susceptibility to pneumococcal infection, and one of the lines in particular, named as mutant94, showed a notably decreased survival of 28% after *S. pneumoniae* challenge, compared to the final survival of 74% (p<0.0001) in challenged wild type (AB) controls (**Fig 3A**). For a comparison, **Fig 3A** shows the survival rates also for the other two hypersusceptible mutant lines, mutant450 (38% survival, p<0.0001), and mutant14 (45% survival, p<0.0001), as well as for two mutant lines with comparable survival to AB larvae, mutant445 (83% survival) and mutant12 (79% survival).

To determine whether the hypersusceptible mutants had defects in resistance or in tolerance, we measured the bacterial burden in the infected larvae at 18 hpi. As shown in **Fig 3B**, the mutant94 larvae had elevated bacterial burden compared to AB larvae (p<0.001) and thus showed decreased ability to control bacterial growth. In contrast, bacterial burden in the other hypersusceptible lines did not differ from the bacterial burden in the wild type larvae. These results indicate that the poor survival of mutant94 larvae after pneumococcal challenge is mainly due to compromised resistance against pneumococcal infection while the mutant450 and mutant14 larvae rather have poorer tolerance.

## Poor *crp2* expression is associated with severe pneumococcal infection in mutant94 larvae

To obtain insight into the underlying defect in the innate defense mechanisms to pneumococcus in the hypersusceptible mutant line, we determined the differentially expressed genes in mutant94 and AB larvae at 18 hours post *S. pneumoniae* challenge by RNA sequencing. Of the 61 above-mentioned protein-coding genes induced in pneumococcal infection, 9 genes were differentially expressed in mutant94 larvae compared to the AB larvae (**S5 Table**). All the genes were downregulated by at least 3-fold and represented the functional categories of

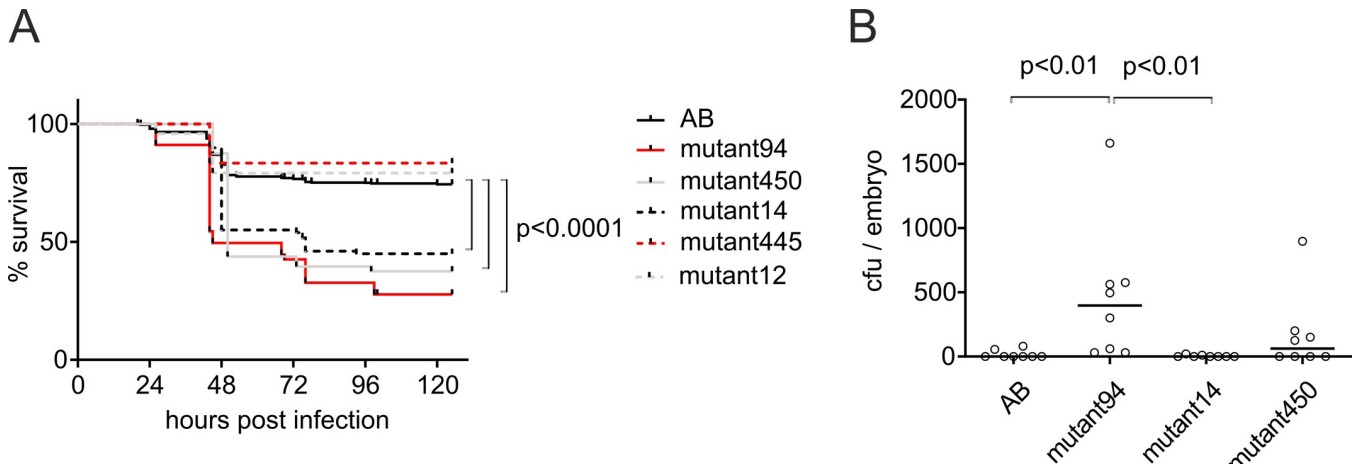

**Fig 3. Mutant94 larvae are hypersusceptible to pneumococcal infection.** A) Survival of three mutant zebrafish lines, mutant94, mutant450, and mutant14, is compromised in pneumococcal infection compared to wild type (AB) fish and other mutant lines (mutant445 and mutant12). In the experiment, mutant and AB embryos were infected with 35–440 cfu of *S. pneumoniae* at 2 dpf. The graphs represent the results from three (mutant94, mutant450, mutant14, AB) or one survival experiment (mutant445 and mutant12) (n = 24–48 in each). B) Bacterial burden in mutant94 larvae is elevated after pneumococcal challenge compared to AB and other hypersusceptible mutant lines. Embryos were infected with 505 cfu of *S. pneumoniae* at 2 dpf and bacterial counts were determined at 18 hpi. The circles represent bacterial counts in a single larva from one experiment (n = 8). Median bacterial count is depicted with a line. The statistical comparison of difference was calculated with log-rank (Mantel-Cox) test in A, and with Kruskal-Wallis test with Dunn's multiple comparisons test in B. cfu = colony forming units.

immune response (4 genes), metabolic processes (2 genes), and unknown processes (3 genes). The biggest difference was observed for the expression of *crp2* (ENSDARG00000056498), with an 83-fold reduction compared to the AB larvae. The second and third greatest reductions in response among protein-coding genes were seen with *BX548011.1* (43-fold decrease in expression*)*, and *si:dkey-9c18.3* (29-fold decrease in expression), for which putative human homologs or functional predictions were not found. In addition, there were more subtle changes (from 3.5- to 7.1-fold reduction) in the expression levels of some other genes involved in immune response (*chia.1, ly6m2, chia.2)* and for genes with other functions (*cyp7a1, zgc:173443, ctrb.3*). Of the observed differences, the poor induction of *crp2* expression in mutant94 appeared to be a plausible explanation for compromised resistance and thus was selected for further analysis. As there are two genes named as *crp2* in CRCz11 reference genome, we will hereafter refer to the one with reduced expression in mutant94 larvae (with the gene ID of ENSDARG00000056498) as *crp2-1*, and the other (with the gene ID of ENSDARG00000056462) as *crp2-2*.

In human hepatocytes, the expression of *CRP* is induced as a part of an acute response to inflammation by the pro-inflammatory cytokines IL1B, IL6 and TNF [77–80]. To examine whether the reduction of *crp2-1* expression in mutant94 larvae was due to a defect in the expression of these mediators, we looked at their expression levels in the mutant94 larvae. As seen in the **Fig 4**, according to the RNA sequencing data, *il1b*, *il6*, and *tnfa* are upregulated during pneumococcal infection in AB (fold-change of 4.4, 9.9, and 2.7 for *il1b*, *il6*, and *tnfa*, respectively, compared to unchallenged larvae), and even more so in mutant94 larvae (fold-change of 3.3, 2.0, and 3.3 for *il1b*, *il6*, and *tnfa*, respectively, compared to challenged AB larvae). Similar induction in *il1b* and *tnfa* expression in pneumococcal infection in AB and mutant94 larvae, as well as reduction in *crp2-1* expression in mutan94 larvae was also seen with quantitative PCR (qPCR), validating the RNA sequencing results (**S1 Fig**). From these results, we concluded that the lack of *crp2-1* expression in the mutant larvae does not result from the general defect in the inflammatory response, and the increase in the expression of

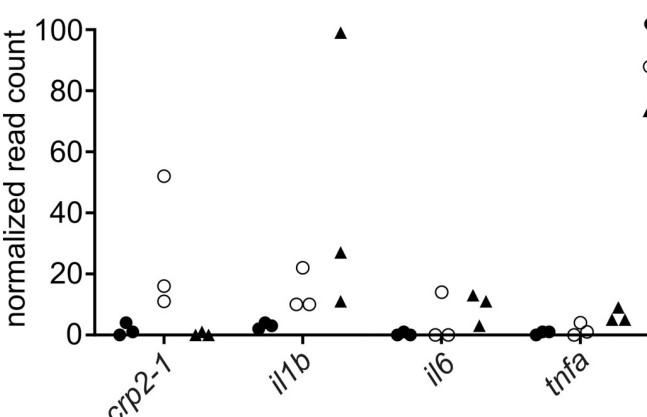

**Fig 4. Mutant94 larvae show hampered induction of *crp2-1* upon pneumococcal infection compared to wild type larvae.** The figure shows normalized read counts for *crp2-1*, *il1b*, *il6*, and *tnfa* in wild type larvae injected with KCl (unchallenged AB), wild type larvae infected with ~500 cfu of *S. pneumoniae* (challenged AB), and mutant94 larvae infected with ~500 cfu of *S. pneumoniae* (challenged mutant94) at 18 hpi. Each dot represents the normalized read count in a sample of 10 larvae. The differential expressions did not reach statistical significance.

*il1b*, *il6*, and *tnfa* in the mutant larvae may be due to a compensatory response to the hampered *crp2-1* expression, or due to more severe infection and higher bacterial burden in mutant94 larvae compared to AB larvae.

## Knockout of *crp2-1* causes a moderate phenotype during pneumococcal infection, and the effect is augmented by concomitant knockdown of *crp3*

According to the current reference genome GRCz11, the zebrafish has six *CRP* homologs, *crp1* (ENSDARG00000071454), *crp2-1* (*crp2*, ENSDARG00000056498), *crp2-2* (*crp2*, ENSDARG00000056462*) crp3 (*ENSDARG00000042613), *crp6* (ENSDARG00000071457*), and crp7* (ENSDARG00000071456) [51]. Of these, the two *crp2* genes and *crp3* are highly similar in sequence (88–89% similarity between *crp2-1*, *crp2-2* and *crp3*), while *crp1*, *crp6*, and *crp7* are somewhat more distinct (81–83% similarity with *crp2-1*, *crp2-2 and crp3*) (analyzed with the BLAST tool of Ensembl Genome Browser [51]). According to the mRNA sequencing data, *crp2-1* and *crp3* have similar expression patterns in our infection model, with induction in challenged AB larvae (fold increase of 14.8 and 3.4 for *crp2-1* and *crp3*, respectively, compared to unchallenged AB) but reduced expression in challenged mutant94 larvae (fold reduction of 83.2 and 2.5 for *crp2-1* and *crp3*, respectively, compared to challenged AB) (**S2 Fig**). *crp2-2*, on the other hand, exhibits opposite expression pattern with no expression in challenged AB larvae but high expression in challenged mutant94 (fold increase of 7.2 compared to challenged AB). According to the data, *crp1* is expressed independent of the treatment, while *crp6* and *crp7* do not seem to be expressed in zebrafish larvae at 18 hpi. Therefore, it seems that *crp2-1*, *crp2-2* and *crp3* may be relevant in pneumococcal infection in zebrafish larvae, and due to the sequence homology and clearly related expression patterns of these three Crp-encoding genes, it is possible, that their products also have overlapping functions.

To gain more insight into the importance of zebrafish Crps in the defense against pneumococcal infection in zebrafish, we produced a *crp2-1* knockout zebrafish line with CRISPR-Cas9 mutagenesis (**Fig 5**). During the past few years, the engineered type II CRISPR/Cas system has become a popular tool for reverse genetics and has been widely used in various animals, including zebrafish, to model human genetic disorders [81,82]. To ensure efficient mutagenesis of *crp2-1* in zebrafish, the sequence of the potential target site for the mutagenesis was

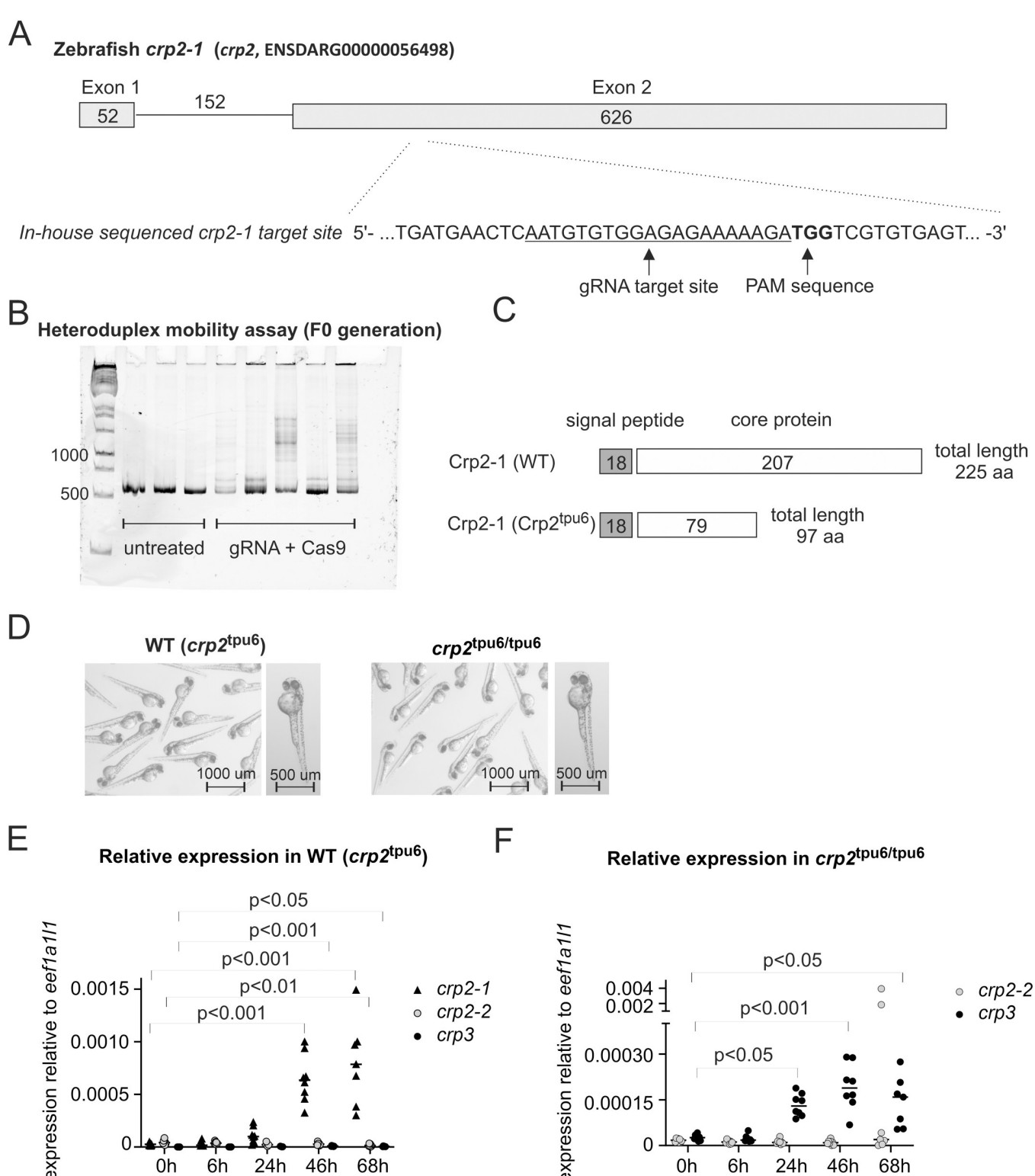

**Fig 5. crp2-1 knockout zebrafish produce truncated form of Crp2-1 and show compensatory expression of crp3.** A) A schematic presentation of the zebrafish *crp2-1* (*crp2*, ENSDARG00000056498) gene structure and the gRNA target site for CRISPR-Cas9 mutagenesis. The target site sequence with depicted gRNA target sequence and protospacer adjacent motif (PAM) site were verified with Sanger sequencing. B) Successful mutagenesis in the embryos injected with target specific gRNA and Cas9 protein was verified with heteroduplex mobility assay. In the heteroduplex mobility assay, multiple bands on polyacrylamide gel after heteroduplex formation indicate the presence of insertion/deletion mutations at the target site. C) Schematic representation of the

structure of wild type (WT) Crp2-1 protein and the structure of Crp2$^{tpu6}$ mutant protein after introduction of +23 nucleotide insertion and premature stop codon by CRIPSR-Cas9 mutagenesis. D) Homozygous *crp2*$^{tpu6/tpu6}$ larvae have normal morphology under standard laboratory conditions. The panel shows an overview and a representative image of a single larvae of 2 dpf wild type (WT (*crp2*$^{tpu6}$)) and homozygous (*crp2*$^{tpu6/tpu6}$) larvae of F3 generation. Images were taken with Lumar V.12 fluorescence stereomicroscope with an exposure time of 100 ms. E) *crp2-1* expression is induced in WT (*crp2*$^{tpu6}$) larvae at 24 hpi. F) *crp3* expression is induced in homozygous *crp2*$^{tpu6/tpu6}$ larvae at 24 hpi. In E and F, the expression relative to *eef1a1l1* expression was measured with the $2^{-\Delta Ct}$ method from the pools of five WT or mutant larvae infected with 241 cfu of *S. pneumoniae*. The line depicts the median expression level, and the statistical analyses were conducted with Kruskal-Wallis test with Dunn's multiple comparisons test. Comparisons were only done to 0 h time point of the same gene.

verified by sequencing and found to be slightly different from the reference sequence in GRCz11 (in-house sequenced wild type sequence presented in **Fig 5A**). By injecting Cas9 protein and a single guide RNA targeting the second exon of *crp2-1*, we induced insertion/deletion mutations in the founder (F0) generation (**Fig 5B**). By sequencing the mutated loci in outcrossed F1 larvae (F0 x AB), we identified several germline-transmitted mutations, one of which, an insertion of 23 nucleotides (loss of A and gain of TGGTCTCCACACAGATGGT CGTGT) at the target locus was chosen for the production of F2 and further generations. This mutation leads to a disrupted reading frame and a premature stop codon after 97 amino acids (aa) in the Crp2-1 protein normally with total length of 225 aa (**Fig 5C**). Of note, homozygous mutants appeared phenotypically normal, with no defects in morphology or well-being in standard laboratory conditions (**Fig 5D**). This knockout mutant line is hereafter called *crp2*$^{tpu6}$.

In addition to the changes in protein structure and function, an indel mutation introduced by CRISPR-Cas9 may also affect the transcript levels of the target gene [83,84]. Multiple recent studies have also demonstrated changes in the expression levels of other, closely related genes in knockout models due to transcriptional adaptation (reviewed in El-Brolosy & Stainier, 2017) [85]. To see whether the mutation affects the expression of *crp2-1* and the related *crp2-2* and *crp3*, we measured the relative expression levels of these genes in homozygous and WT larvae. For the analysis, *crp2*$^{tpu6/tpu6}$ and WT (*crp2*$^{tpu6}$) larvae of F3 generation were infected with 241 cfu of *S. pneumoniae* at 2 dpf and the infected larvae were collected at 0h, 6h, 24h, 46h, and 68h for the analysis of gene expression by qPCR. Due to the high level of homology between the three Crp-encoding genes, the specificity of the amplification in selected *crp2*$^{tpu6/tpu6}$ and WT (*crp2*$^{tpu6}$) samples was evaluated by Sanger sequencing. Consistent with the analysis of *crp2-1* expression in challenged AB larvae, *crp2-1* expression was induced in challenged WT (*crp2*$^{tpu6}$) larvae after 24 hpi by 3.9-fold (p<0.01) (**Fig 5E**). *crp2-2* and *crp3*, on the other hand, only showed minor expression in the same larvae and no induction upon pneumococcal challenge (**Fig 5E**). We were unable to verify the presence or absence of *crp2-1* transcript in homozygote mutants by qPCR due to the amplification of transcripts aligning equally well with *crp2-1*, *crp2-2* and *crp3* (**S3 Fig**). Sequencing of the amplified cDNA of *crp2*$^{tpu6/tpu6}$ revealed overlapping spectra, but through manual interpretation, we identified one of the sequences as *crp2-1* with the +23 nt mutation. Of note, no such amplification of multiple targets was detected in WT (*crp2*$^{tpu6}$) larvae. The expression levels of *crp2-2* and *crp3* in the challenged *crp2*$^{tpu6/tpu6}$ larvae showed only minor expression and no induction of *crp2-2* upon pneumococcal infection, but clear, 5-fold induction of *crp3* after 24 hpi (p<0.01) (**Fig 5F**). Overall, while the unspecific amplification in homozygote samples but not in WT samples might suggest a decrease in the level of *crp2-1* transcript in the mutants, this could not be verified with the qPCR analysis. However, these results show that whether present at equal or lower levels than in WTs, the *crp2-1* transcripts in *crp2*$^{tpu6/tpu6}$ mutants carry the specific frame-shifting insertion and the premature stop codon leading to a truncated Crp2-1 protein in the mutant larvae. In addition, the increased *crp3* transcript levels in *crp2*$^{tpu6/tpu6}$ larvae suggest activation of compensating mechanisms for the loss of functional *crp2-1* transcript or protein.

Next, to see whether the phenotype observed in mutant94 larvae could be explained by Crp2-1 deficiency, we infected homozygous, heterozygous and WT *crp2*$^{tpu6}$ embryos at 2 dpf

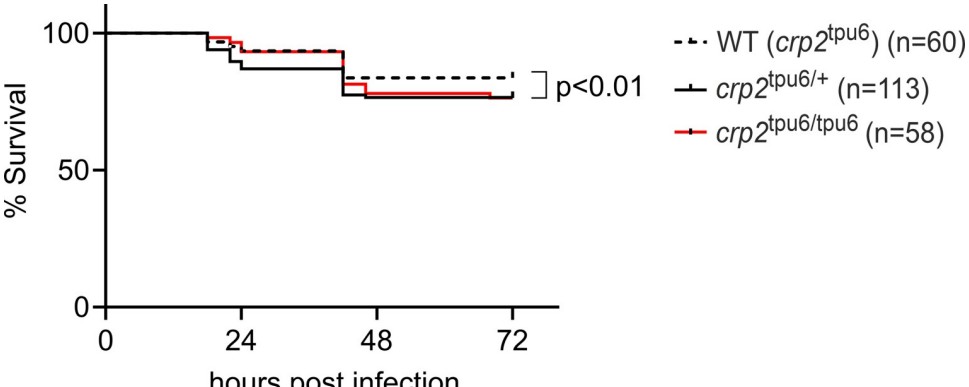

**Fig 6. crp2<sup>tpu6</sup> mutants show modest impairment in survival in systemic pneumococcal infection.** The figure shows the survival of wild type (WT (crp2<sup>tpu6</sup>)), heterozygous (crp2<sup>tpu6/+</sup>) and homozygous (crp2<sup>tpu6/tpu6</sup>) larvae after pneumococcal infection. In the experiment, 1,646 cfu of *S. pneumoniae* was injected into the bloodstream of 2 dpf larvae and the survival was monitored until 5 dpf. Data were collected from a single experiment with the group sizes (n) indicated in the figure. The statistical comparison of difference was calculated with log-rank (Mantel-Cox) test.

with 1,646 cfu of *S. pneumoniae* and monitored their survival for three days. The experimental infections and the survival assay were carried out in a blinded and randomized setting as described in *Materials and Methods*. As shown in **Fig 6**, both the homozygous and heterozygous mutants showed decreased survival compared to their WT siblings with survival percents of 76%, 77% and 84% for *crp2<sup>tpu6/tpu6</sup>*, *crp2<sup>tpu6/+</sup>*, and WT (*crp2<sup>tpu6</sup>*), respectively (p<0.01). While the modestly compromised survival in the mutants indicates that Crp2-1 does participate in the defense response to pneumococcus in this model, the milder phenotype of the knockout mutant compared to the mutant94 indicates that other factors also contribute to the hypersusceptibility of mutant94 observed in the forward genetic screen. Alternatively, as was suggested by the increased expression of *crp3* in *crp2<sup>tpu6/tpu6</sup>* larvae, it is plausible that the observed transcriptional adaptation to the loss of function of Crp2-1 results in a milder phenotype in pneumococcal infection model.

To test whether *crp3* compensates Crp2-1 deficiency, we used morpholino-oligonucleotides to knockdown *crp3* expression in *crp2<sup>tpu6/tpu6</sup>* larvae. A splice-blocking (SB) morpholino targeting the *crp3* 5' upstream sequence and tentatively causing the skipping of exon 1 where the start codon lies, was chosen for the study. Using a dose titration test, dose of 4 ng per embryo was chosen as this was the maximum dose that showed no developmental effects within the first 5 dpf in unchallenged AB larvae (**Fig 7A**). To address the knockdown efficiency of *crp3* SB morpholino, the amount of WT *crp3* transcript in morpholino injected larvae was determined by qPCR utilizing primers which tentatively bind only to WT transcript. As a sign of knockdown, the *crp3* SB morphants showed lower levels of WT *crp3* transcript at 3–4 dpi, the most important time considering *S. pneumoniae* infection kinetics in zebrafish larvae, with residual expression of 24% at 3 dpi (p<0.01) and 48% at 4 dpi (p<0.05) compared to larvae injected with the same dose of random control (RC) morpholino (**Fig 7B**). To test the synergistic effect of *crp2-1* knockout and *crp3* knockdown, *crp3* SB morpholino and RC morpholino injected *crp2<sup>tpu6/tpu6</sup>* larvae were infected with 296 cfu of *S. pneumoniae* at 2 dpf and their survival was followed for three days. During the infection experiment, no signs of off-target effects affecting development could be detected in either of the groups. The survival of *crp3* SB injected larvae from pneumococcal infection, on the other hand, was significantly lower (53% survival) compared to the RC injected larvae (75% survival) (p<0.05). We therefore conclude, that while the knockout of *crp2-1* alone seems to have only a moderate effect on zebrafish

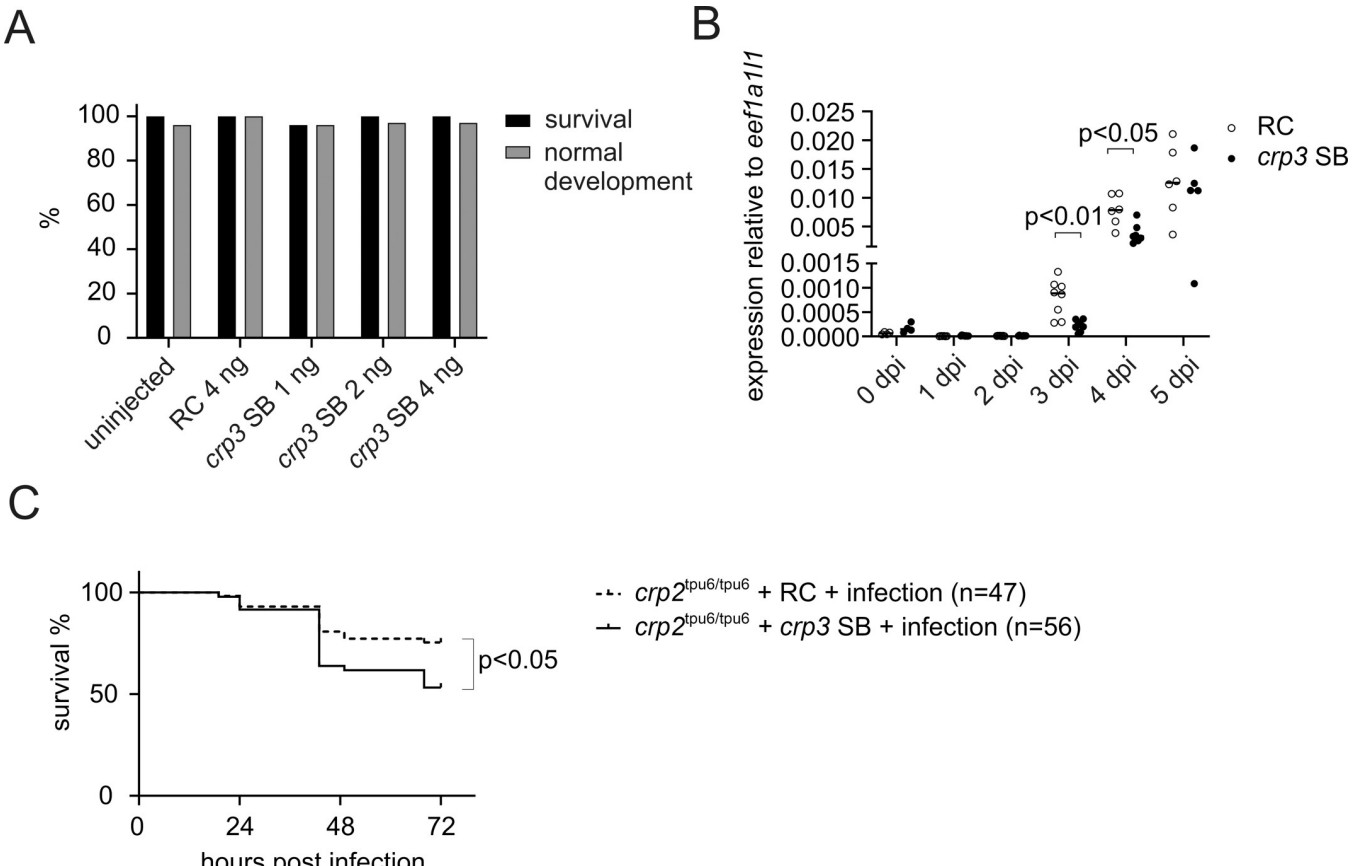

**Fig 7. Partial knockdown of *crp3* further impairs the survival of *crp2*<sup>tpu6/tpu6</sup> larvae from systemic pneumococcal infection.** A) Percentage of surviving and normally developing uninjected zebrafish larvae, larvae injected with 4 ng of RC morpholino, and larvae injected with different doses (1 ng, 2 ng or 4 ng) of *crp3* SB morpholino. In the experiment, RC or *crp3* SB morpholino was microinjected into the yolk sac of AB zebrafish embryos at the 1–4 cell stage and their survival and development was followed for 5 days. The percentage represents the live (black) and morphologically normal (gray) larvae in each group (n = 31–50) at 5 dpi. B) *crp3* transcript levels in RC morpholino and *crp3* SB morpholino injected AB zebrafish larvae at 0–5 dpi. In the experiment, 4 ng of RC or *crp3* SB morpholino was microinjected into the yolk sac of AB zebrafish embryos at 1–4 cell stage and pools of 3–5 larvae (n = 7–8) were collected at specific time points. *crp3* expression levels were measured with qPCR from technical duplicates and normalized with *eef1a1l1* expression. A dot/circle represents the relative gene expression in a single pool of embryos/larvae, which was calculated using the $2^{-\Delta Ct}$ method, and the line depicts the median expression level. C) The survival of RC or *crp3* SB morpholino injected homozygous (*crp2*<sup>tpu6/tpu6</sup>) larvae after pneumococcal infection. In the experiment, 4 ng of RC or *crp3* SB morpholino was microinjected into the yolk sac of *crp2*<sup>tpu6/tpu6</sup> embryos of F3 generation at 1–4 cell stage, and at 2 dpi the larvae were infected with 296 cfu of *S. pneumoniae*. The survival of infected larvae was monitored until 5 dpf. Data were collected from a single experiment, with the group sizes (n) indicated in the figure. The statistical comparisons of difference were calculated with Kruskal-Wallis test with Dunn's multiple comparison in B, and with log-rank (Mantel-Cox) test in C. dpi = days post injection.

survival from systemic pneumococcal infection, together with *crp3* knockdown it results in significantly compromised survival of the larvae. However, this phenotype still does not fully recapitulate the phenotype of mutant94 larvae, indicating that other genes, such as *BX548011.1* which was the second most downregulated gene in mutant94 larvae, or *crp2-2*, may affect the resistance against *S. pneumoniae* in zebrafish larvae. This however, remained to be experimentally investigated as we failed to knockdown the expression of *BX548011.1* and *crp2-2* in zebrafish larvae by specific SB morpholino oligonucleotides.

## Discussion

It is widely known that individuals with weakened or immature adaptive immune response are more prone to severe pneumococcal infection. This includes the very young, the very old, and

those with genetic or acquired immunodeficiencies, for example, due to defects in antibody production, HIV infection, or transplantation [20,86]. While antibodies against capsule polysaccharide and protein antigens are the main components of protective immunity to pneumococcus, defects in innate mechanisms, especially the complement system and TLR signaling, have also been associated with increased susceptibility to invasive pneumococcal infection [20,87,88]. Complement activation and the subsequent opsonophagocytosis, in particular, are critical in the clearance of pneumococcus, as has been shown in multiple knockout studies in murine models of pneumococcal infection [89–96]. Similarly, a deficiency in the production of complement components has been associated with an increased risk for several bacterial (including pneumococcal) infections in humans [87,97–100]. Supporting the importance of complement-mediated clearance of pneumococci, the virulence of pneumococcus seems to be at least in part linked to their ability to resist complement [101]. In fact, many pneumococcal virulence factors have been reported to interact with the complement components and to disturb various steps in the proteolytic cascade [102–109]. These include, for example, the capsule, pneumolysin, and Autolysin A, which, according to the previous studies by us and others, are also important determinants of pneumococcal virulence in zebrafish [28,39,110].

Although much of the functional characterization of the zebrafish immune system remains to be done, the current knowledge on the subject suggests a high level of conservation between the immune systems of fish and humans. In general, zebrafish possess all the cell types of the human immune system [111–120], as well as counterparts for human Toll-like receptors, NOD-like receptors, and the components of the related signaling cascades [44,121–124]. Moreover, counterparts for almost all the complement components have been described in zebrafish [42,125]. With the aid of reverse genetic approaches, transcriptome analyses, and *in vivo* imaging of pathogens and immune cells, the host-pathogen interactions in several bacterial infections in zebrafish have been shown to resemble those in mammals [28,44,45,126–129]. In our study, similar conservation was observed in the host response to pneumococcal infection. We used the previously established zebrafish larval model for the systemic pneumococcal infection and characterized the innate immune response to pneumococcus at 18 hpi. The transcriptome analysis showed that, as in humans, pneumococcus triggers an inflammatory response in zebrafish larvae, characterized by the increased expression of genes coding for pro-inflammatory cytokines, chemokines, antimicrobial peptides, and acute phase proteins. We have previously reported that the intravenous injection of pneumococcus activates the migration of myeloid cells to the site of infection and leads to the phagocytosis of the bacteria and that the depletion of myeloid cells significantly hampers zebrafish survival from the pneumococcal infection [28]. This was also supported by the present data, which revealed the increased expression of genes associated with leukocyte migration and phagocytosis. The importance of phagocytic cells in the clearance of pneumococci in zebrafish is also suggested by the number of upregulated genes coding for known or postulated opsonins, including the complement components, members of the zebrafish Crp multiprotein family, and Itln3. In humans, CRP promotes the phagocytosis of *S. pneumoniae* either through the activation of the classical pathway of complement or through direct binding to the Fc gamma receptor on macrophages and dendritic cells [130–132]. The ability of human intelectins to recognize specific sugar moieties in the cell wall of pneumococcus, on the other hand, is a relatively new discovery, and their exact roles in the host defense remain to be elucidated [63,64]. Supporting our results, zebrafish *intelectin 3* is highly induced upon bacterial infection, and the functional characterization of this protein family in zebrafish and other species has shown the ability of intelectins to agglutinate bacteria and promote phagocytosis [84,133–136].

The complement system of zebrafish has not been functionally characterized and whether it serves an analogous role in the innate immune response as in mammals, is not known.

However, counterparts for all the activation pathways have been described in zebrafish and the complement related genes have also been shown to be upregulated in several bacterial infections [126,127,137,138]. Similarly, the complement related genes were clearly enriched in our transcriptome data, suggesting their important roles in the early response to pneumococcus also in this model. According to the transcriptome data, in particular, the classical and alternative pathways may participate in the response to pneumococcus in zebrafish, since the infection induced the expression of genes specifically associated with these pathways. This includes the genes coding for the counterparts for human CRP known to activate the classical pathway upon pneumococcal infection in humans, as well as the counterparts for C1R and CFB required for the activation of C3 convertase in the classical and the alternative pathways, respectively, and C4B, a component in both the C3 and C5 convertase complexes. All the three pathways converge at the level of component C3, whose production was also upregulated in zebrafish larvae upon pneumococcal infection. Unlike humans, who possess only one *C3* gene, the zebrafish genome codes for eight C3 variants [139]. Six of the zebrafish C3 variants (coded by *c3a.1-6*) exhibit significant homology to human C3, and have been shown to respond to bacterial stimulus, while the other two (coded by *c3a.7*, *and c3a.8*) are more distinct, both in sequence, and in expression pattern [139]. The most responsive C3 genes in our infection model were *c3a.1-3* and *c3a.6*, supporting the previous study and further suggesting their important role in host defense during bacterial infection. Whether the proteins encoded by these genes are analogous in function is still under investigation.

The high resemblance of the innate immune response to pneumococcus in zebrafish to that in humans gives a solid basis for the use of zebrafish larvae for further characterization of the host response to pneumococcus. We therefore performed a medium-scale forward genetic screen for novel genes associated with the innate immune response to pneumococcus. The screen identified one mutant line with an increased bacterial burden and decreased survival in systemic pneumococcal infection. The determination of differentially expressed genes in the mutant and wild type larvae revealed that the expression of *crp2-1 (crp2,* ENSDARG00000056498*)* was decreased by 83-fold in the mutant larvae compared to the wild type larvae. Human CRP is an acute phase protein which can bind the C polysaccharide in the pneumococcal cell wall and activate the classical pathway of complement [94,140]. CRP bound to pneumococcus can also directly stimulate macrophages and dendritic cells and promote the intake of bacteria and the expression of pro-inflammatory cytokines [131,132]. While polymorphism in the CRP-encoding gene seems to occur rarely, some studies have shown association between polymorphism and susceptibility to pneumococcal infection in humans [22,26,141]. Studies in mice also indicate an important role for CRP as a part of the host defense against pneumococcus as *CRP* knockout mice show increased mortality after pneumococcal challenge [142]. In addition, treatment with human CRP prior to pneumococcal challenge has been shown to protect mice from severe pneumococcal bacteremia, and while the exact mechanisms behind the protection are not known, it is thought to be related to complement activation [143–147].

Originally, seven homologs (*crp1-7*) of human *CRP* were identified in zebrafish [148], while in the GRCz11 genome assembly, only six homologs (*crp1*, *crp2-1*, *crp2-2*, *crp3*, *crp6* and *crp7*) are currently listed. The analysis at the amino acid level shows that all the zebrafish Crps show equal similarity (31–34%) to the human CRP (Comparisons done by Clustal Omega [149,150]). Despite this relatively low-level similarity, the overall structure, including the ligand binding pocket as well as binding sites for C1q and Fc receptor, is relatively well conserved in zebrafish [151–153]. As the functionality of the zebrafish Crps has not been analyzed in detail, it is not possible to assign functional homology, although the existence of this protein family across species and its retained ability to activate complement in fish suggests that it

serves a similar function in zebrafish [154,155]. Of note, the Crp multi-protein family in zebrafish exhibits notable heterogeneity with, not only, multiple isoforms with potentially different ligand specificities [152], but also, multiple transcript variants observed for Crp coding genes. As suggested for other diversified gene families, such as those coding for innate receptors [121] and complement components [42], the existence of such heterogeneity may provide wider pathogen recognition in the aquatic environment and at the same time compensate the more modest adaptive responses in these lower vertebrates.

As in humans, at least some of the zebrafish Crp isoforms seem to be acute phase proteins since their expression is induced upon bacterial as well as viral infections [138,156,157]. In the case of *crp2-1*, this was also evident in our data. Zebrafish isoforms, especially Crp2-2, Crp3, and Crp6, have also been shown to have antiviral activity in Spring viraemia of carp virus (SVCV) infection in zebrafish by modulating autophagy process and the level of reactive oxygen species [156,158]. Consistent with previous studies, our expression analyses show a heterogenous response of Crp isoforms to bacterial infection, with the isoforms encoded by *crp2-1*, *crp2-2* and *crp3* displaying relevance to the anti-pneumococcal response. At the same time, the expression analyses in wild type and mutant lines suggest overlapping functions for these three isoforms. Most importantly, the compromised survival in zebrafish lines with either diminished *crp2-1* expression (mutant94) or mutated *crp2-1* (*crp2*tpu6) suggest an essential role of zebrafish C-reactive protein in anti-pneumococcal immunity. Furthermore, when the *crp2-1* knockout is combined with partial knockdown of *crp3*, the effect on survival is elevated which further supports their important role in zebrafish immune response and indicates overlapping functions of different Crp isoforms. Whether zebrafish Crps also act through conserved effector functions, however, is yet to be unraveled.

As a summary, our data indicate that the human pathogen pneumococcus activates an innate immune response in zebrafish analogous to that in mammals and highlights the importance of complement-mediated immunity in the anti-pneumococcal response. Phenotypic analyses of two zebrafish lines with defects in the expression of a gene homologous for human *CRP* suggest a conserved role also for this pathway in fish. Our data, therefore, give evidence on the high-level conservation of the innate immune response to this human pathogen, and reveal a set of genes with a previously unrecognized function in the host defense against pneumococcus for further functional characterization. The characterized expression patterns of the genes encoding for zebrafish Crp isoforms in wild type and *crp2-1* knockout larvae also give valuable information on the heterogenous activities of zebrafish Crp isoforms, whose functional characterization is only beginning. Finally, the transcription data show multiple pneumococcus-responsive genes with no clear homolog in mammals, and whose function is yet to be determined. Once characterized, these genes might provide important insights into the evolution of vertebrate immune system.

## Materials and methods

### Zebrafish and ethics statement

The wild type zebrafish (*Danio rerio*) lines were obtained from and maintained by the Tampere Zebrafish Core Facility (Tampere university, Finland). The AB wild type zebrafish line was used as a reference line in the infection experiments and in the production of the mutant lines. The TL (Tupfel long fin, *leo*t1 -/-, *lof*dt2 -/-) line was used in the outcrosses in the production of the mutant lines for the forward genetic screen. Zebrafish were maintained according to the standard protocols [159] and the ethical guidelines set by the Ethical Board in Finland. Briefly, zebrafish embryos and larvae were kept in E3-medium (5 mM NaCl, 0.17 mM KCl, 0.33 mM $CaCl_2$, 0.33 mM $MgSO_4$, 0.0003 g/l Methylene Blue) in an incubator at 28˚C, with a

light/dark cycle of 14/10 hours until 6 dpf after which they were moved to a filtered flow-through system with the same light/dark cycle and fed once a day with GEMMA Micro 75–500 (Skretting, Stavanger, Norway) or twice a day with SDS 100–400 (Special Diets Services, Essex, UK). The well-being of the fish was monitored daily, and moribund fish were immediately euthanized with an overdose (0.08%) of an anesthetic Tricaine (3-aminobenzoic acid ethyl ester, pH 7.0) (Sigma-Aldrich, St. Louis, Missouri, USA). During the experiments, embryos and larvae were maintained in corresponding conditions. Prior to the treatments, zebrafish larvae were anesthetized with 0.02% Tricaine. The maintenance and the experiments were done in accordance with the Finnish Act on the Protection of Animals Used for Scientific or Educational Purposes (497/2013) and the EU Directive 2010/63/EU. The ethical permissions for the generation of the mutant zebrafish lines as well as the maintenance of the fish were granted by the Animal Experiment Board in Finland (the licenses ESAVI/10079/04.10.06/2015, ESAVI/6407/04.10.2012, ESAVI/10366/04.10.07/2016, ESAVI/2235/04.10.07/2015, ESAVI/11133/04.10.07/2017, ESAVI/2464/04.10.07/2017, ESAVI/2776 /2019 and ESAVI/7251/2021). All the zebrafish experiments were conducted prior to the independently feeding larval stage and therefore, according to the EU directive, no permission for experimentation was required.

### The gene-breaking tol2 transposon-based mutagenesis and the generation of mutant fish lines for the forward genetic screen

Generation of the zebrafish mutant lines for the forward genetic screen was carried out with the method published by Balciunas et al (2006). and Clark et al. (2011) [76,160], and as described in detail in Harjula et al. (2020) [45]. Briefly, the gene-breaking transposon pGBT-RP2.1 (RP2) vector (Addgene plasmid # 31828) and the pT3TS-Tol2 vector coding for the tol2 transposase (Addgene plasmid # 31831) [76] were transformed into *E. coli* One Shot TOP10 cells (Invitrogen, Thermo Fisher Scientific, Waltham, Massachusetts, USA) and purified from overnight cultures with the QIAGEN Plasmid Plus Maxi Kit (Qiagen, Hilden, Germany). To produce *tol2* mRNA, the linearized pT3TS-Tol2 plasmid was used as a template in *in vitro* transcription conducted with the mMESSAGE mMACHINE T3 Kit, (Invitrogen, Thermo Fisher Scientific) according to the manufacturer's instructions. To carry out the mutagenesis, 12.5 pg RP2 and 12.5 pg *tol2* mRNA in phosphate buffered saline (PBS) and with 0.6% phenol red (Sigma-Aldrich) were microinjected into the 1-cell stage AB zebrafish embryos and the mutation carrying embryos (designated as F0) were selected based on the expression of *green fluorescent protein (GFP)* under the Lumar V.12 fluorescence stereomicroscope (Carl Zeiss MicroImaging GmbH, Göttingen, Germany). To produce the mutant zebrafish lines, F0 founder fish were outcrossed with TL, resulting in the F1 generation of fish potentially carrying multiple insertional mutations. These fish were named after a running number and, again, were crossed to TL to obtain F2 generation fish. The F3 and the following generations (until F5) were generated by incrosses. In some cases, in order to maintain the line, the fish were again crossed to TL. In each generation, mutation-carrying progeny were selected based on the expression of *GFP*.

### CRISPR-Cas9 mutagenesis

CRISPR-Cas9 mutagenesis in zebrafish was carried out based on the methodology by Hruscha and Schmid (2015) [161]. An optimal target sequence for a guide RNA (gRNA) targeting *crp2-1* was selected with an online CRISPR design tool CHOPCHOP v2 [162], after which the sequence was manually adjusted to match with the target sequence in our AB variant. NCBI BLAST (https://blast.ncbi.nlm.nih.gov/Blast.cgi) analysis against the zebrafish reference

genome GRCz11 was used to verify the specificity of the chosen target sequence 5'-AATGTGTGGAGAGAAAAAGATGG-3'. The production of gRNA and the mutagenesis were carried out as previously described in Uusi-Mäkelä et al. (2018) [163] with a few adjustments. Briefly, a gRNA targeting the second exons of *crp2-1* was produced from a DNA oligo template (Sigma-Aldrich) with the MEGAshortscript T7 Transcription Kit (Invitrogen, Thermo Fisher Scientific) according to the manufacturer's instructions. For the mutagenesis, a mixture of 230 pg of the gRNA, 330 pg of in-house produced Cas9 protein (Protein Service core facility, Tampere University, Finland) and 1% Rhodamine dextran tracer (Invitrogen, Thermo Fisher Scientific) was injected into the cytoplasm of 1-cell stage AB zebrafish embryos and the mutagenesis efficiency was evaluated by heteroduplex mobility assay (HMA) [163] from isolated DNA of 2 dpf embryos. Germline-transmitted mutations at the target site were screened in the progeny of outcrossed F0 zebrafish using HMA and Sanger sequencing, and F1 zebrafish with a desired nonsense mutation were incrossed to generate a stable knockout zebrafish line.

## Genotyping of CRISPR mutants

At all stages of the mutant line production, genotyping of the adult fish was carried out from the tailfin DNA by Sanger sequencing. Specifically, fish were anesthetized with 0.02% Tricaine, and a small piece of tail fin was cut off with a scalpel blade. Tail fin tissue was lyzed for 4–24 h at 55˚C in lysis buffer (10mM Tris pH 8,2, 10mM EDTA, 200mM NaCl, 0.5% SDS, 200μg/ml Proteinase K). DNA was precipitated for 1h-24h at -20˚C using two volumes of ethanol after which DNA was pelleted by centrifuging at 16,000g for 10min. The pellet was washed with 70% ethanol and resuspended in water. The genomic site containing the mutagenesis target site was amplified using DreamTaq Hot Start DNA Polymerase (Thermo Scientific) and sequenced using the BigDye Terminator v3.1 Sanger sequencing Kit (Applied Biosystems, Life technologies, Carslbad, USA). Primers used in the genotyping are listed in **S7 Table.** To distinguish homozygous and wild type fish, the obtained sequence was compared to the wild type sequence using Clustal Omega [149,150]. To analyze the mutation site in the heterozygote fish, CRISP-ID v1.1 web application [164] for detecting CRISPR induced indels was used. Alternatively, the two overlapping sequences (one from the WT allele and the other from the mutant allele) were manually separated from the sequencing chromatogram.

## Generation of zebrafish morphants and quantification of knockdown efficiency

A splice-blocking morpholino against the zebrafish *crp3* was designed by and ordered from GeneTools LLC (Philomath, OR, USA) and its specificity for *crp3* was checked with NCBI BLAST tool (https://blast.ncbi.nlm.nih.gov/Blast.cgi). The *crp3* SB morpholino oligo sequence was 5'-TGCTCCAAGTCTGAAAAGGAAGAAA-3'. To determine the maximum dose of the morpholino with no off-target effects, three doses (4 ng, 2 ng and 1 ng) of the *crp3* SB morpholino were microinjected into the yolk sac of 1–4 cell stage wild type AB zebrafish embryos. Prior to microinjection, morpholinos were heated at 65˚C for 10 min to ensure complete suspension. In microinjection, injection volume of 1 nl per embryo was used and the injection solution was dyed with 1% Rhodamine dextran tracer (Invitrogen, Thermo Fisher Scientific) to verify the success of the injection. After injection, the development and well-being of injected larvae were monitored daily for five days, with abnormal and dead embryos/larvae recorded.

To quantify the effect of *crp3* SB morpholino on the levels of WT *crp3* transcript, AB zebrafish embryos were microinjected as above with 4 ng of *crp3* SB morpholino or random control

(RC) morpholino (GeneTools LLC) and pools of 3–5 larvae were collected daily from 0 to 5 dpi for the extraction of total RNA.

### Experimental *S. pneumoniae* infection and survival assay

The *Streptococcus pneumoniae* wild type strain TIGR4 (T4) of serotype 4 was used throughout the study [165] and the cultivation and the harvesting of the bacteria was carried out as previously reported [28]. Briefly, bacteria were grown overnight on 5% lamb blood agar plates (Tammer BioLab Ltd, Tampere, Finland) at 37˚C and 5% $CO_2$ and after that, in liquid culture in Todd Hewitt broth (Becton, Dickinson and Company, New Jersey, USA) supplemented with 0.5% yeast extract (Becton, Dickinson and Company) until mid-logarithmic growth phase determined by optical density at a wavelength of 620 nm. Bacterial cells were harvested by centrifuging for 10 min at 4000 rpm (rounds per minute). Using the theoretical bacterial density in the liquid culture ($10^8$ cells/ml), the bacterial pellet was suspended in desired volumes of 0.2 M KCl (infection experiments related to the genetic screen and mutant94) or PBS (infection experiments related to *crp2*[tpu6]) to obtain the desired bacterial dose. The actual number of bacteria in the injection solution was verified by plating samples of single injections on 5% lamb blood agar plates (Tammer BioLab Ltd). Due to variation in bacterial growth rate, accuracy of absorbance measurement, slightly varying volumes of bacterial solution released by the injector, accuracy of quantitative plating etc., relatively big variation in bacterial doses between experiments (30–570 cfu in infection experiments related to the genetic screen and mutant94 and 296–1,646 cfu in infection experiments related to *crp2*[tpu6]) was seen. To avoid the influence of varying doses, all groups within a single experiment were infected with the same bacterial solution and only groups from the same assay were compared to find differences in survival or bacterial count.

Experimental *S. pneumoniae* infections and the subsequent survival assays were carried out by microinjecting a specific dose of *S. pneumoniae* into the blood circulation valley of 2 dpf zebrafish larvae and following the survival for 72–96 hpi as described in detail in Rounioja et al. (2012) [28]. The unchallenged control groups were similarly treated with either 0.2 M KCl (infection experiments related to the genetic screen and mutant94) or PBS (infection experiments related to *crp2*[tpu6]). The survival assay for the *crp2*[tpu6] mutant line was carried out in F2 larvae (the progeny of incrossed F1 heterozygotes) showing Mendelian ratios of homozygous, heterozygous, and wild type larvae. To achieve a blinded and randomized experimental set up, the larvae were infected prior to the genotyping. In the experiment, the survival of the infected larvae was followed for 72 hours, during which the larvae were checked twice a day. Terminally diseased and dead larvae were collected, euthanized if necessary, and frozen for later genotyping. For the genotyping, the genomic DNA extraction from whole larvae, target site amplification, and sequencing, were done as described for adult fish genotyping in the section *Genotyping of CRISPR mutants*.

### Determining the bacterial load from zebrafish larvae

The bacterial loads were measured from infected zebrafish larvae at desired time points by quantitative plating of homogenates. More specifically, infected larvae were euthanized with 0.08% Tricaine (Sigma-Aldrich) and washed three times with PBS. By using a pestle, single larvae were manually homogenized in 200 μl of PBS with 1% Triton-X (Sigma-Aldrich). To obtain the bacterial counts, the homogenates were serially diluted and grown on 5% lamb blood agar plates (containing 1 μg/ml erythromycin) (Tammer BioLab Ltd) overnight at 37˚C and 5% $CO_2$.

## RNA extraction and the quality control

For the RNA sequencing, total RNA was extracted from the pools of five larvae using the Qiagen RNeasy Mini Kit (Qiagen) according to the manufacturer's instructions. The genomic DNA was removed from the samples using the RapidOut DNA removal Kit (Thermo Scientific, Thermo Fisher Scientific) and the purity and the concentration of the samples were measured with Qubit RNA BR Assay Kit (Invitrogen, Thermo Fisher Scientific). The RNA integrity was analyzed with the Fragment Analyzer (Advanced Analytical Technologies, Iowa, USA) using the Standard Sensitivity RNA Analysis Kit (Advanced Analytical Technologies) and the PRO-Size® 2.0 Data Analysis Software (Advanced Analytical Technologies). The RNA samples with an integrity number (RIN) >9 were used in the transcriptome analysis. Finally, two RNA samples were combined to achieve sufficient concentration for the transcriptome analysis.

For the qPCR analysis of *crp2-1*, *crp2-2* and *crp3* expression in WT and mutant *crp2*[tpu6] larvae and of *crp3* expression in *crp3* SB morpholino and RC morpholino injected larvae, total RNA was extracted from the pools of 3–5 larvae using the Qiagen RNeasy Plus Mini Kit with genomic DNA elimination step (Qiagen) according to the manufacturer's instructions. The purity and the concentration of the samples were measured with NanoDrop 2000 Spectrophotometer (Thermo Scientific, Thermo Fisher Scientific).

## Quantitation of the relative gene expression levels by qPCR

The expression levels of *il1b*, *il6*, *tnfa and crp2-1* in the RNA sequencing samples were measured with qPCR using the iScript One-Step RT-PCR Kit with SYBR® Green (Bio-Rad Laboratories, Hercules, California, USA) according to the manufacturer's instructions. Thermal cycling was performed using the Bio-Rad CFX 96 instrument (Bio-Rad Laboratories) and the CFX Software version 3.1 (Bio-Rad Laboratories) was used to analyze the results. The target gene expression levels were normalized to the *eef1a1l1* expression [166], and the double delta Ct method ($2^{-\Delta\Delta Ct}$) was used to calculate the fold change in expression in challenged larvae compared to unchallenged larvae.

For the measurement of the expression levels of *crp2-1*, *crp2-2* and *crp3* in mutant and wild type *crp2*[tpu6] larvae, and the expression levels of *crp3* in *crp3* SB morpholino and RC morpholino injected larvae, the reverse transcription of the RNA samples was done with the SensiFast cDNA synthesis kit (BioLine, London, UK) and the qPCR with the PowerUp SYBR® master mix (Applied Biosystems, Thermo Fisher Scientific). Thermal cycling was performed using the Bio-Rad CFX Opus 96 instrument (Bio-Rad Laboratories) and the CFX Maestro Software version 2.2 (Bio-Rad Laboratories) was used to analyze the results. The specificity of the amplified qPCR products was evaluated by melt curve analysis and Sanger sequencing of selected samples. Delta Ct method ($2^{-\Delta Ct}$) was used for calculating the expression levels of target genes relative to the expression of *eef1a1l1*. The primers used for the qPCR are listed in **S7 Table**.

## RNA sequencing and data analysis

The cDNA library preparation and the RNA sequencing were conducted at Novogene, Hong Kong. The 250–300 bp cDNA library was sequenced with the 150 bp paired-end sequencing on the Illumina platform and with a sequencing depth of >20 million reads/sample.

The quality of the reads was inspected using the FastQC and the reads were aligned against the GRCz10 reference genome with STAR using the default parameters (http://www.bioinformatics.babraham.ac.uk/projects/fastqc) [167]. To improve the alignment accuracy, Ensembl GRCz10.91 was used as the reference set for known transcripts. FeatureCounts was used to quantify the expressions of the genes belonging to the same reference gene set as that used during the alignment step [168]. Normalization of raw expressions and differential gene expression analysis were both conducted using R-package DESeq2 [169].

## Statistical analyses

The ClinCalc web tool (https://clincalc.com/Stats/SampleSize.aspx) was used to carry out the calculations for sample sizes in survival experiments. Based on our previous experiments in *S. pneumoniae* challenged zebrafish larvae, the difference in end-point mortality between wild type and mutant lines was estimated to be 40%. With an 80% statistical power, a minimum group size of 16 was determined for survival experiments.

All the statistical analyses were carried out with the GraphPad Prism software version 5.02 or 9.0.0 (GraphPad Software, Inc, California, USA). The log-rank (Mantel-Cox) test was used to analyze the differences in the survival rates between the zebrafish lines. The differences in the gene expression levels measured by qPCR and the bacterial amounts were analyzed with Kruskal-Wallis with Dunn's multiple comparisons test in Figs **3B**, **5E–5F** and **7B** and with two-tailed Mann-Whitney test in **S1 Fig**. In each analysis, a p-value of <0.05 was considered as statistically significant and only p-values <0.05 are shown in the figures.

## Supporting information

**S1 Table. Induced protein coding genes not previously reported in the context of immune response.**
(PDF)

**S2 Table. Downregulated protein coding genes in pneumococcal infection.**
(PDF)

**S3 Table. Non-coding RNAs upregulated in pneumococcal infection.**
(PDF)

**S4 Table. Downregulated non-coding RNAs in pneumococcal infection.**
(PDF)

**S5 Table. Downregulated protein coding genes in mutant94 larvae.**
(PDF)

**S6 Table. Downregulated non-coding RNAs in mutant94 larvae.**
(PDF)

**S7 Table. Primer sequences.**
(PDF)

**S1 Fig. Relative expression of *il1b, il6, tnfa, and crp2-1* in AB and mutant94 larvae.**
(PDF)

**S2 Fig. Expression of zebrafish Crp-encoding genes in AB and mutant94 larvae 18 hours post pneumococcal challenge.**
(PDF)

**S3 Fig. Alignment of qPCR product sequences from homozygous *crp2^{tpu6/tpu6}* mutants with *crp2-1*, *crp2-2*, and *crp3* reference sequences.**
(PDF)

## Acknowledgments

We would like to thank Leena Mäkinen, Jenna Ilomäki, Tuula Myllymäki, Markus Ojanen, Essi Mäkinen, Jenni Jouppila, Heather Mathie, and Nicholas Halfpenny for technical assistance and Matthew Maasdorp for proofreading the manuscript. We warmly thank Professor

Stephen C. Ekker's laboratory (Mayo Clinic, Rochester, USA) for a generous gift of gene-breaking transposon pGBT-RP2.1 (RP2) vector and the pT3TS-Tol2 vector. The authors also acknowledge the Tampere Zebrafish Laboratory, the Biocenter Finland (BF) and Tampere Genomics Facility, and the Tampere facility for protein services (PS) for their service.

## Author Contributions

**Conceptualization:** Anni K. Saralahti, Sanna-Kaisa E. Harjula, Olli Lohi, Samuli Rounioja, Mataleena Parikka, Mika Rämet.

**Formal analysis:** Anni K. Saralahti, Tommi Rantapero.

**Funding acquisition:** Anni K. Saralahti, Sanna-Kaisa E. Harjula, Mika Rämet.

**Investigation:** Anni K. Saralahti, Sanna-Kaisa E. Harjula, Meri I. E. Uusi-Mäkelä, Mikko Kaasinen, Maiju Junno, Hannaleena Piippo.

**Methodology:** Anni K. Saralahti, Sanna-Kaisa E. Harjula, Mataleena Parikka, Mika Rämet.

**Resources:** Mataleena Parikka, Mika Rämet.

**Supervision:** Anni K. Saralahti, Matti Nykter, Olli Lohi, Mataleena Parikka, Mika Rämet.

**Visualization:** Anni K. Saralahti.

**Writing – original draft:** Anni K. Saralahti.

**Writing – review & editing:** Sanna-Kaisa E. Harjula, Tommi Rantapero, Meri I. E. Uusi-Mäkelä, Mikko Kaasinen, Maiju Junno, Hannaleena Piippo, Matti Nykter, Olli Lohi, Samuli Rounioja, Mataleena Parikka, Mika Rämet.

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
