## [Decision Letter · Decision Letter 0]

12 Jul 2022

Dear Dr Rämet,

Thank you very much for submitting your Research Article entitled 'Characterization of the innate immune response to Streptococcus pneumoniae infection in zebrafish' to PLOS Genetics.

The manuscript was fully evaluated at the editorial level and by independent peer reviewers. The reviewers appreciated the attention to an important problem, but raised some substantial concerns about the current manuscript. Based on the reviews, we will not be able to accept this version of the manuscript, but we would be willing to review a much-revised version. We cannot, of course, promise publication at that time.

The editors strongly encourage you to perform the additional experiments suggested by Reviewer 1 for further consideration of the manuscript.

If you decide to revise the manuscript for further consideration at PLOS Genetics, please aim to resubmit within the next 60 days, unless it will take extra time to address the concerns of the reviewers, in which case we would appreciate an expected resubmission date by email to plosgenetics@plos.org.

[LINK]

We are sorry that we cannot be more positive about your manuscript at this stage. Please do not hesitate to contact us if you have any concerns or questions.

Yours sincerely,

Man-Wah Tan, Ph.D

Associate Editor

PLOS Genetics

Hua Tang

Section Editor: Human Variation

PLOS Genetics

Reviewer's Responses to Questions

**Comments to the Authors:**

Reviewer #1: see attachment

Reviewer #2: Summary

The paper by Saralahti et al. describes how the innate immune system of zebrafish larvae is affected by Streptococcus pneumoniae infection and highlights how the mechanisms involved are conserved between zebrafish and human. To this end, they used an established infection model of zebrafish embryos and performed a whole-genome transcriptomic analysis to identify the altered pathways following pneumococcal infection. They found protein coding genes as well as non-coding RNAs presenting a differential gene expression level, some were upregulated, and others were downregulated. Most of these genes were associated with the immune response, but some were involved in metabolic pathways or had unknown functions.

Most importantly, a comparison to the human innate immunity-related genes expressed during pneumococcal infection showed well conserved mechanisms, notably components of the complement and in particular the complement activating protein CRP - for which two homologs exist in the zebrafish (crp2, crp3), had their expression altered.

Next, the authors further validated their observations through a forward genetic approach where they screened 126 generated zebrafish mutant lines for susceptibility to pneumococcal infection: they ended up with one mutant (mutant94) that showed exceptionally low resistance to infection compared to AB control and other mutant lines. Data from RNA-sequencing performed on this mutant revealed an important impairment of crp2_1 gene expression (a human CRP homolog).

To further confirm the previous results and investigate the mechanisms in more detail, the authors decided to generate a crp2_1 knockout zebrafish line using CRISPR-Cas9, resulting in a truncated Crp2-1 protein, and proceeded with the same pneumococcal infection protocol. Here again, they observed a poorer survival rate of the knockout mutant line compared to AB control (although less dramatic than what they observed in mutant94), which confirmed the importance of the C-Reactive Protein in immunity against pneumococcal infection.

Although the importance of the innate immune system following infection in zebrafish has been highlighted in previous studies (Saralahti et al. 2014), the high-throughput approach used here brought a global view that also allowed the discovery of new potential effectors previously unknown to be involved in the response to pneumococcal infection in zebrafish larvae. Because evidence of the conservation of the mechanisms related to innate immunity between zebrafish and human was shown, the results also suggest a possible conservation of the other pathways, possibly in humans, which emphasizes the high relevance of the use of the zebrafish model for this type of studies.

The authors introduced the public health issue very nicely making the relevance of the model and the study very convincing. The story was described in a clear manner, easy to follow and understand, with simple but robust and very well executed approaches as well as the use of a vast array of relevant techniques. I think the authors proved their point with a nicely designed study.

Minor points

Results

Line 213 to 215: I first thought the phrase was unfinished and a verb was missing at the end but then realized the misuse of “comprised of”, which is the case also on line 241 and can make these sentences difficult to understand.

I think “be” should be added to make it into the verbal expression “to be comprised of xyz” or “of” needs to be removed if the verb “to comprise xyz” is used instead.

The 2 paragraphs from line 207 to 233: I think one should refer to Table 1 as the last figure mentioned was Fig2, so it can get confusing since everything discussed is about Table1.

Another general comment I have is regarding the placement of the citations: in some cases (for example line 223 to 233), the citations are left at the end of the sentence even though it ends with a description of the authors’ own results from what I understood. I believe the citation in such examples should be moved higher up, right after the cited part, to avoid confusions.

Lines 248-250: very impressive massive work!

Lines 252 – 253 : 30-570 cfu, Why such a big range of CFU? Were different CFUs tested or was it just meant to be 300-570?

261 to 267: The resistance vs tolerance experiment. I get the point and it’s very well explained, but I’m wondering, even though it is quite clear that mutant94 has impaired resistance, couldn’t it also be that in addition to that it also has impaired tolerance? (because they also die more than the other hypersusceptible lines according to Fig 3A) I don’t think this requires any further experimental proof, but just a point maybe to add in the interpretation of the data?

Line 363: why more than 1000 cfu are used?

I am also curious as to why different cfu were used throughout the study? (500 cfu in Fig 2, 300 cfu in Fig3 and 1600 cfu in fig6). I don’t recall this point being addressed, but my apologies if I missed it.

Line 370 - Although an in-depth characterization of the truncated protein in the KO mutant might be complicated and probably unnecessary here, maybe on the other hand a further genome analysis of the mutant94, sequencing, genotyping, might be of interest to actually identify the other components potentially involved together with crp2_1 and impaired in this mutant?

But I understand these approaches can be complicated and time-consuming, when in my opinion the authors proved their point already in this paper, but it could be an interesting follow-up for further studies maybe.

Fig 4: very clear and to the point results, but just curious, why was the unchallenged mutant not included?

Discussion

Very nicely discussed topic, with a clear take-home message, I like the unexpected opening to a whole new field (evolution of vertebrate immune system)!

**Have all data underlying the figures and results presented in the manuscript been provided?**

Reviewer #1: Yes

Reviewer #2: Yes

PLOS authors have the option to publish the peer review history of their article (what does this mean?). If published, this will include your full peer review and any attached files.

Reviewer #1: **Yes: **Jacqueline M Kimmey

Reviewer #2: **Yes: **Peter Bergman

---

## [Decision Letter · Decision Letter 1]

20 Dec 2022

Dear Dr Rämet,

We are pleased to inform you that your manuscript entitled "Characterization of the innate immune response to Streptococcus pneumoniae infection in zebrafish" has been editorially accepted for publication in PLOS Genetics, pending some minor revisions as noted below. Given that you did attempt but failed to knockdown crp2-2 and BX548011.1, and that crp-3 SB morpholino in combination with crp2-1 KO still did not fully recapitulate the phenotype arising from the screen, it will be important to state those attempts in the manuscript and note that you cannot rule out the contributions of crp2-2 and BX548011.1 to the phenotypes you assessed.  

Yours sincerely,

Man-Wah Tan, Ph.D

Academic Editor

PLOS Genetics

Hua Tang

Section Editor

PLOS Genetics

Comments from the reviewers (if applicable):

Reviewer's Responses to Questions

**Comments to the Authors:**

Reviewer #2: All my questions have been addressed in an adequate manner. Thanks!

**Have all data underlying the figures and results presented in the manuscript been provided?**

Reviewer #2: Yes

PLOS authors have the option to publish the peer review history of their article (what does this mean?). If published, this will include your full peer review and any attached files.

Reviewer #2: **Yes: **Peter Bergman, MD, PhD, Professor

**Data Deposition**

http://datadryad.org/submit?journalID=pgenetics&manu=PGENETICS-D-22-00563R1

**Press Queries**

---

## [Editor Report · Acceptance letter]

4 Jan 2023

PGENETICS-D-22-00563R1 

Characterization of the innate immune response to Streptococcus pneumoniae infection in zebrafish 

Dear Dr Rämet, 

We are pleased to inform you that your manuscript entitled "Characterization of the innate immune response to Streptococcus pneumoniae infection in zebrafish" has been formally accepted for publication in PLOS Genetics! Your manuscript is now with our production department and you will be notified of the publication date in due course.

With kind regards,

Zsofi Zombor

PLOS Genetics

On behalf of:
